# Arterial Presumed Perinatal Ischemic Stroke: A Mini Review and Case Report of Cognitive and Speech-Language Profiles in a 5-Year-Old Girl

**DOI:** 10.3390/children11010033

**Published:** 2023-12-28

**Authors:** Ivana Bogavac, Ljiljana Jeličić, Maša Marisavljević, Tatjana Bošković Matić, Miško Subotić

**Affiliations:** 1Cognitive Neuroscience Department, Research and Development Institute “Life Activities Advancement Institute”, 11000 Belgrade, Serbia; i.bogavac@add-for-life.com (I.B.); m.marisavljevic@add-for-life.com (M.M.); m.subotic@add-for-life.com (M.S.); 2Department of Speech, Language and Hearing Sciences, Institute for Experimental Phonetics and Speech Pathology, 11000 Belgrade, Serbia; 3Department of Neurology, Faculty of Medical Sciences, University of Kragujevac, 34000 Kragujevac, Serbia; boskovic-matic.tatjana@medf.kg.ac.rs; 4Clinic of Neurology, University Clinical Centre of Kragujevac, 34000 Kragujevac, Serbia

**Keywords:** arterial presumed perinatal ischemic stroke, cognition, speech-language development, neuroplasticity

## Abstract

Arterial presumed perinatal ischemic stroke is a type of perinatal stroke that emerges due to late or delayed diagnostics of perinatal or neonatal arterial ischemic stroke. It is usually recognized before one year of life due to hemiparesis. This injury may lead to cognitive, behavioral, or motor symptoms, and life-long neurodevelopmental disabilities. In this case report, we describe a five-year-old girl with a history of arterial presumed perinatal ischemic stroke in the left hemisphere, which adversely affected her cognitive and language outcomes. The girl’s cognitive development has been uneven, ranging from below average to average, and she had specific language acquisition deficits in comprehension, vocabulary, morphology, use of complex syntax, and narrative structure. The obtained results point to the specificity of each child whose development is influenced not only by the timing of the brain lesion and the degree of damage, but also by the child’s neurobiological capacity. In addition, we provide an updated review of the literature that includes information on epidemiology, risk factors, diagnostics, clinical manifestations, outcomes, and potential therapies. The present article highlights the importance of early intervention and systematic monitoring of children with perinatal stroke with the aim of improving the child’s development.

## 1. Introduction

Perinatal stroke, even though it is a rarity, leads to a heterogeneous group of neurological injuries that occur in both fetuses and neonates, as well as in infants and children. Many of these injuries are symptomatic and can be detected at the early stages; consequently, they are a cause of lifelong neurodevelopmental difficulties. The term perinatal stroke covers a group of cerebrovascular diseases with poor outcomes and patients with lifelong neurodevelopmental difficulties/disabilities. It is recognized as the most common cause of unilateral spastic cerebral palsy (USCP), while many patients have epilepsy, persisting intellectual disabilities, and developmental and behavioural disorders [1]. Perinatal stroke can be divided according to cerebrovascular events into haemorrhage vs. ischemia [2]. The classification diversity arises from the absence of the exact timing of the cerebrovascular event and its clinical manifestations. In some cases, symptomatology is present during the neonatal period, while in others, they may be unrecognized or undiagnosed because symptoms do not indicate neuroimaging. Perinatal haemorrhagic stroke is an intracerebral haemorrhage that affects parenchymal, intraventricular, and leptomeningeal areas [3]. Other types of lesions, such as cerebral sinovenous thrombosis (CSVT) and periventricular haemorrhagic infractions (PVHI), will not be discussed in this paper. Perinatal ischemic stroke refers to a group of conditions with disrupted cerebral blood flow between 18 gestational weeks and 28 postnatal days. Neuroimaging improvements have led to the distinction of different subcategories of perinatal ischaemic strokes based to lesions, risk factors, and pathophysiology, in addition to the age of symptom onset and clinical presentations. The terminology of arterial ischemic stroke, which is determined by the time of its occurrence, is shown in Figure 1.

The cerebrovascular event in the prenatal period, from 18 gestational weeks to 7 postnatal days, is covered with the terms perinatal arterial ischemic stroke (PAIS), perinatal arterial stroke (PAS), and perinatal ischemic stroke (PIS). PAIS is present in term and preterm-born infants [4,5,6]. In the neonatal period, from birth to 28 postnatal days, the cerebrovascular event is commonly marked as neonatal arterial ischemic stroke (NAIS) [3,5,6]. It is significant to distinguish these two terms because NAIS may be used instead of PAIS, despite having different ethology and clinical manifestations. The group of arterial strokes diagnosed later, beyond 28 postnatal days, in the first few months of life and usually in infants not thought to be neurologically ill, are stated as arterial presumed perinatal ischemic stroke (APPIS). The diagnosis is usually set after the appearance of reach and grasp asymmetry, seizures, or failure to reach developmental milestones [5,6].

### 1.1. Epidemiology and Pathophysiology

Explaining incidence and risk factors is challenging due to methodological differences (hospital-based/population studies, preterm/term born children, different diagnostic codes). Recent studies report that the incidence depends on the child’s age; the incidence is higher in newborns (1/3500 live births) than childhood (1–2/100,000 per year). Incidence peaks are presented in the perinatal period (5–13/100,000 live births), in children under 5 years of age (0.38/100,000 per year), and in adolescence (0.48–0.6/100,000 per year) [7]. Data about sexes indicates a male predominance [8,9,10], while 15% are preterm children [2]. The associated mortality of perinatal stroke is very low and usually results from additional complications: massive haemorrhage, total hypoxia/ischemia, or other systematic conditions. The pathophysiology of arterial ischaemic stroke is, to some extent, unexplained, and mainly arises from individual cases. The well-known fact is that PAIS is a consequence of arterial occlusion, but how and why occlusion happens is unclear [2]. Ischemic brain injury after thrombus or embolus is followed by a cascade of events with cell death. The placenta-embolic mechanism is the most recognized for occlusion, wherein a thrombo-embolus from the placenta passes through the patent foramen ovale into the cerebral circulation [2,11].

### 1.2. Etiology and Risk Factors of Perinatal Stroke

The etiology is most likely multifactorial, with an increased risk if multiple factors [2,12] and risk factors specific to pregnancy [13] are present. It is important to emphasize that risk factors are not the same as the cause of prenatal stroke. Many studies have shown its multifactorial nature because of hemostasis changes during pregnancy and the complexities of maternal-fetal circulations [10,12,13,14,15,16,17,18]. All the described risk factors can be classified into maternal, antenatal, intrapartum, and fetal factors (Table 1).

Maternal factors also include infertility [13,14], history of stillbirth [15], infection/fever [13,14,18], a prolonged second stage [13], and abdominal pain [10]. Some authors have included an arterial umbilical cord pH < 7.10 [10,18] among the intrapartum risk factors.

### 1.3. Clinical Manifestations and Diagnostics of Perinatal Stroke

The timing of stroke occurrence is essential for clinical manifestations and diagnostics. The stroke, in utero, usually will not have acute manifestations. It will most likely be unnoticeable if it is followed by regular or minimal complications at birth and no symptoms in the neonatal period. Additional testing is needed for adequate diagnosis with no symptoms or subtle ones. APPIS is diagnosed in infants without neurological symptoms during the first months of life. The first sign is usually hemiparesis, occurring between four and eight months old [19,20], becoming evident during voluntary motor activity, such as using arms to grab an object [5,7,20]. This asymmetry leads to early handedness or fisting, and in 81–86% of cases, is the most common complaint of parents [14,21]. With time, these features become more noticeable and are usually diagnosed as USCP. In addition to this asymmetry, global developmental delay, numbness or weakness, and post-neonatal seizures can indicate prenatal arterial injuries [7,14,20,21]. These symptoms are late manifestations of early injury, and magnetic resonance imaging (MRI) is the gold standard for obtaining diagnosis [2,7,12,14,22]. In order to obtain optimal imaging modalities, defining the extent of an injury, the tissue involved, and properly predicting outcomes and possible therapies should be done for any child with suspicion of stroke. Important sequences include “diffusion-weighted imaging (DWI), T_1_- and T_2_-weighted imaging (T_1_W and T_2_W), and susceptibility weighted imaging” [14]. According to literature findings on NAIS [2,23], T_1_W and T_2_W images in infants older than 1–2 months show tissue atrophy or cysts in the infracted area. APPIS is easily recognizable because it involves an arterial vascular territory, and in the majority of cases, is unilateral, with the left hemisphere lateralized within the middle cerebral artery (MCA) territory [8,12,18]. Accordingly, APPIS is likely to have similar imaging representations and follow the same outcome patterns [6].

### 1.4. Outcomes

Results from recent risk factors studies of presumed perinatal ischemic strokes (PPIS) [24] suggest that arterial prenatal ischemic strokes, regardless of timing (neonatal or presumed), are often a representation of the same disease. APPIS, in most cases, include cortical structures, so late-term outcomes in the form of seizures and developmental delays (language, cognitive, behavior) can be the first symptoms and, in most cases, do not necessitate MRI. Consequently, APPIS outcomes are moderately severe compared to other perinatal stroke outcomes [1]. In the absence of longitudinal, follow-up studies, the 50–75% percentage of infants with long-term sequelae diagnosed in the perinatal period [20] is probably higher, including patients with developmental delays that emerge later in infancy. It must be taken into consideration that, in most cases, the diagnosis is given during the first year of life while the brain functions are still developing, and possible deficits may arise later. For this reason, the neurological development of infants diagnosed with APPIS must be monitored to prevent and adequately treat newly emerging deficits.

Regardless of the time of onset, outcomes mainly depend on the localization and size of the stroke. In the literature, the most common consequence of APPIS is USCP with extremely high rates [6,12], post-neonatal seizures (40–50%) [4,6,12], while cognitive, language, and behavioral problems emerge later in childhood. Data are uneven due to the different durations of follow-up studies.

Motor deficits are the most common sequelae in children with APPIS, affecting both arms and legs in different patterns depending on the size and location of the stroke. Motor functions of an arm are often more damaged, which is the representation of the most involved area, the MCA. According to the literature, almost all children will start walking independently between 14 and 18 months of age [8,25]. To some extent, motor outcomes can be predicted if scans show involvement of the basal ganglia, while nonmotor outcomes result from cortical involvement [14]. The second common outcome is post-neonatal seizures. The incidence of epilepsy is inconstant depending on the timing and duration of follow-up studies. At least one reappearance of seizures is observed in 38–50% of children [2,4,6,26], while 11–13% had active epilepsy [25]. Children with seizures at birth or spasms are more likely to develop epilepsy later in childhood [26].

Neurodevelopmental deficits include deficits in cognition, language, behavior, and other higher brain functions [1], with numerous inconsistencies. The complexity of development itself, the timing of deficit onsets, different measures, different ages of children at the time of testing, durations of follow-up studies, and small samples, all affect the descriptions and consistency of results. Considering everything, most children’s global intelligence quotient (IQ) fall within the normal range, and these children attend regular schools [27,28,29,30]. Lower IQ scores are more present in children with epilepsy or those experiencing high-frequency and severe postnatal seizures. A very important feature for the interpretation of results is the age of children testing. Many studies have shown a decrease in IQ scores as children get older [31,32,33,34], but this is not always the case, and it could be explained in a few ways. The frequency and severity of post-neonatal seizures affects cognitive outcome, leading to decreased scores and altered developmental courses [35,36]. The presence of seizures may influence learning and the acquisition of knowledge by causing new functional damage; consequently, children may have lower IQ scores. Attention deficits can affect or contribute to executive function problems [37]. It could also be explained by insufficient brain plasticity; more complex brain functions cannot be accommodated, causing deficits to emerge [35]. Scarce data are available on executive functions (behavioral and emotional self-regulation) in the population of children with APPIS. There is evidence of lower average scores for attention, inhibitory control, processing speed, flexibility and shifting, planning, and organization [6]. Some patterns are observed in spatial abilities regarding the localization of a stroke. Right hemisphere lesions induce more global errors with shape or design, in the way that children have difficulties in putting elements together to make a whole, while left hemisphere lesions induce more local errors—recall of details [27]. This pattern is observed in adults where spatial cognition is localized in the right hemisphere.

Clinical studies have shown the remarkable ability of the developing brain to reorganize, especially when it comes to language skills. Although massive left-sided lesions are present, language skills have a tendency to be well-preserved. Nevertheless, deficits and difficulties are reported. Right-sided injuries have more impact on the comprehension of basic vocabulary and nonverbal communication (gestures, for example), while initial vocabulary production deficits are more present in left-sided lesions. The latest controlled functional magnetic resonance imaging (fMRI) study [38], which included only participants with PAIS to the MCA of the left hemisphere, with significant infractions in language areas, focused on language tasks in older children and young adults. The study showed that the frontotemporal cortex of the right hemisphere is responsible for sentence processing, with no differences between the control and experimental groups. The same study also showed that the right hemisphere regions, which are dominant for emotional prosody processing in healthy individuals, maintain dominance for emotional prosody processing and sentence processing in individuals with PAIS. Other MRI studies that examined perinatal strokes have shown evidence of recovery and reorganization of language abilities in the frontotemporal regions of the right hemisphere [39,40,41,42,43,44]. Despite this functional reorganization, there is evidence of left hemisphere activation and bilateral activation during language tasks [6]. Some authors confirm the incidence of language impairment at the same rates as their peers, 20–25% [6,45], while others report rates up to 49–50% [46,47]. Likewise, cognitive outcomes and language outcomes vary significantly depending on the type of language tasks, study methodology, and participants’ age. More grammatical errors were present in children younger than 7 years old when forming narratives based on pictures. Their stories were shorter, with simplified syntax and a smaller number of details, regardless of having adequate vocabulary. Older children (ages 8 to 10) form narratives with complex syntax and adequate morphology [27].

### 1.5. Follow-Up and Therapy

Children with APPIS are diagnosed months or years after the acute event, with the emergence of one symptom, while others may appear later during childhood. This is the reason no medical treatment is provided, and the focus is on rehabilitation. Early identification is significant, so the adequate stimulation for affected functions can be applied at the right time.

## 2. Case Presentation

### 2.1. Case Report

The girl presented for a detailed cognitive and speech–language assessment at the age of five. She is the third child of young, healthy, nonconsanguineous parents (mother 32, father 35 years). She lives with her parents, older brother, and older sister in a small town in Serbia, and she is a native Serbian speaker from a monolingual family. She was born at term through natural birth, with an Apgar score of 9/10 and a birth weight of 3350 g. At the age of five months, the parents started a medical procedure because they noticed that she did not use her right arm and held her thumb inside her hand. The first examination confirmed semiflexion and hypertonia of the right arm with fisting. After a neurology consultation, she had an MRI scan, and at the age of 6 months, was diagnosed with infractions cerebri (massive perinatal ischemic stroke, predominantly at the left MCA). The neuroimaging presentation of a massive unilateral cortical-subcortical ischemic stroke, dominantly in the left MCA territory, obtained by MRI, is presented in Figure 2.

The child underwent further analysis and was included in physiotherapy when, at this time point, she had monoparesis. The standard electroencephalogram (EEG) findings in the awake state showed spike waves in the left front-central-temporal (FCT) region, while in deep non-rapid eye movement sleep (NREM), changes were grouped as unilateral left-sided paroxysms with post-paroxysmal weakening of the basic activity. All other tests and screenings were neat, and she was heterozygous for MTHFR C677T. At eighteen months, she was diagnosed with USCP, with no data on cognitive or speech and language status. She started to walk at 22 months and had a limited vocabulary with a few meaningful words but no sentences. At this time, the EEG in the awake state and during deep sleep showed a basic activity of about 7 Hz with a superimposed beta rhythm, demonstrating well-formed sleep phenomena over the right hemisphere but not over the left, and without specific graphic elements. Physiotherapy and speech and language therapy were advised.

The girl started speech-language therapy at four years of age at the Institute for Experimental Phonetics and Speech Pathology (IEPSP) in Belgrade. Upon admission, the child was barely verbal, uttering a limited amount of speech. Her speech-language development was reduced, both in terms of comprehension and production. Her comprehension was reduced to simple language instructions and narratives. She used a small number of words (nouns, short words, and a few verbs) to express needs or name objects or actions. Syntactic development was at the level of a holophrase with elementary grammaticality features. However, despite significant deviation in speech-language development, her speech was functional. Therapy was set to maximally exploit her potential and fulfil unique needs during treatment. Individual treatment based on Kostic’s selective auditory filter amplifier (KSAFA) principles was chosen, because it has proven effective in children with complex disorders [48,49].

In summary, the girl’s psychophysiological profile indicates that she experienced delayed psychomotor and speech-language development, leading to USCP after suffering a massive unilateral cortical-subcortical ischemic stroke prenatally, which primarily affected the left MCA territory. This case study is fundamentally based on the clear evidence that language acquisition and other outcomes are affected by perinatal injury. On the other hand, findings regarding the development of children who have had perinatal strokes are inconsistent, with some suggesting that a younger age at the time of stroke is a risk factor for poorer cognitive outcomes [50,51,52,53,54,55], while other findings show that this does not apply linearly to all cognitive outcomes, but is primarily moderated by the location of the lesion [56,57,58]. Therefore, we highlight a few aims of this case report as follows. Firstly, we aim to present the speech-language and cognitive profile of a five-year-old girl with APPIS, primarily in the left MCA territory, after one year of continuous speech-language therapy, to emphasize the uniqueness of every single case, regardless of the time and degree of the insult. In addition, we aim to further clarify the pathways of speech-language and cognition development in this particular group of children, with an emphasis on the specificity of each case influenced not only by the timing of the lesion and the degree of damage, but also by the child’s neurobiological capacity. Finally, but just as importantly, we aim to emphasize the significance of early intervention and the fact that these children need to be included in SLP treatment to minimize the consequences caused by perinatal stroke.

### 2.2. Data Collection

An experienced child psychologist did the cognitive assessment. The girl was tested in a quiet, plain room in the early morning hours. Cognitive skills were tested using the REVISK [59], a revised version of the Wechsler Intelligence Scale for Children in Serbian. According to Wechsler’s principles, the standardized test measures children’s overall intellectual functioning and cognitive abilities. The instrument provides insight into total verbal and performance scores, where higher scores reflect higher levels of intellectual functioning. The Verbal and Performance Scales consist of 5 subtests, with Information, Comprehension, Arithmetic, Similarities, and Digit Span belonging to the Verbal Scale subtests, and Picture Completion, Picture Arrangement, Block Design, Object Assembly, and Coding belonging to the Performance Scale subtests. The total IQ score was not calculated, since the girl’s profile points to uneven development of verbal and performance skills.

Speech-language assessment was done by a speech and language pathologist (SLP) with an MA degree and 18 years of clinical practice. It was conducted in the morning, in a quiet room with no distractions. The girl was seated at a desk, with the SLP sitting opposite. The complete assessment of speech and language included available tests used in clinical practice for decades in Serbia, though not all are standardized. These tests included the Scale for Evaluation of Psychophysiological Abilities of Children (SEPAC) [48,60,61,62], the Peabody Picture Vocabulary Test (PPVT) [63,64], Children’s Grammar Test [60,65], Comic Strip Story [66,67], Global Articulation Test [60,62,68], Test for Analytical Estimation of Serbian Sounds [60], and the Oral Praxis Test [60,62].

The SEPAC is used to estimate psychophysiological development of children in comparison with chronological age represented in months. It includes three subscales—speech and language, sensorimotor, and socioemotional scales. Each scale score is specific for chronological age, and the estimation is done according to the child’s achievements, chronological expectations, and chronological age. For more details, see Rakonjac et al. [62].

The PPVT is designed to estimate receptive vocabulary by requiring the child to point out the picture that represents a named object, action, or phenomenon.

The Children’s Grammar Test aims to assess receptive and expressive knowledge of grammar forms and rules. It estimates the understanding and use of plural, gender, cases, verbal nouns, pronouns, adjectives, verbs, propositions, adverbs, and syntax. The child needs to finish the examiner’s statement based on pictures shown by the examiner. The 42 pictures are designed to elicit certain grammar forms. If the child is not able to produce the adequate grammar form, the examiner then starts to assess grammar form comprehension. For a more detailed description see Bogavac et al. [60].

The Comic Strip Story is the instrument that clinicians use to elicit narratives. It is based on the event sequence represented by four pictures in the form of a comic strip. The child is supposed to tell the story without any model given.

The Global Articulation Test is a screening test for the assessment of production and types of articulation errors in the Serbian phonetic inventory. It consists of 30 words; the child repeats the target word after the examiner, and potential errors are noted. The errors are classified according to articulation characteristics (place and manner) and acoustic characteristics. Afterwards, phonetic transcription (IPA) is used [60,62].

The Test for Analytical Estimation of Serbian Sounds assesses sound production and types of errors regarding specific sound characteristics. It is designed to elicit the child’s production based on a particular picture. The test is designed to elicit a particular sound in the initial, medial, and final positions in the word. Afterwards, every sound is assessed trough pronunciations in the sentence. There are 30 sentences that the child needs to repeat after the examiner. In each sentence, the particular sound is frequently present, which enables assessment during continuous speech. Every aspect of misarticulation is noted and afterwards transcribe with phonetic transcription (IPA) [62].

The Oral Praxis Test assesses the ability to plan and execute orofacial movements after a demonstration by the examiner. It includes 31 oromotor movement of the tongue, jaw, and lips, as well as some adequate posture, velopharyngeal function, voice, and breathing control [60,62].

The child’s parents gave written informed consent for cognitive and speech-language assessment and participation in this study. Humane and compassionate care were provided to the child. In the context of family relationships and actual research, it was negotiated with the parents with the intention of sharing knowledge of the patient’s medical and psychosocial needs and interests. The complete study protocol was in accordance with the Ethical Principles in Medical Research Involving Human Subjects, as established by the Declaration of Helsinki, and had been approved by the Ethics Committee of the Institute for Experimental Phonetics and Speech Pathology in Belgrade, Serbia (No C-22-01; Date: 10 February 2022).

## 3. Results

### 3.1. Cognitive Profile

Table 2 presents information on the child’s cognitive profile. Cognitive assessment results were as follows: verbal intelligence quotient (VIQ)—66 and performance intelligence quotient (PIQ)—93. This cognitive profile points to the uneven development of verbal and performance skills. Scores are also characterized by an uneven scatter within both the verbal as well as performance scales. For instance, within the verbal scale, the girl shows an above-average score on Digit span (+1SD), which indicates good short-term memory and the ability to memorize orally presented information. However, she scored below average on Comprehension (−1SD), which indicates difficulties in understanding more complex questions and tasks and expressing herself verbally.

Skills are also unevenly developed within the performance scale. Above-average scores are seen on Picture Arrangement (+1SD) and Coding (+2SD), which indicates good confluence in visual perceptual ability, thinking and planning, good visual short-term memory, as well as the ability to accomplish a visual motor task within time constraints. On the other hand, she scored below average on Block Design (−2SD) and Object Assembly (−1SD), indicating poor constructive skills. For example, deficits within visuomotor construction show relatively poor abstract problem-solving skills and the ability to analyze an object and construct the whole visual object from its parts within time constraints.

### 3.2. Speech-Language Profile

Our participant showed a very complex speech-language profile. The PPVT showed an age equivalent of six years and one month, and the SEPAC revealed a delay in two scales (Table 3). The speech and language scale was estimated at four years and two months, the sensorimotor scale showed an age equivalent of four years and seven months, while the socio-emotional status was at her chronological level of five years. She was able to understand concrete and abstract questions (e.g., “Can you tell me something about a house?”, “Can you tell me what you should do when you are hungry?”, Can you tell me what you should do when you are cold?”), and have conversations about everyday activities. However, when faced with more complex questions with complex morphosyntactic structures, she was not able to comprehend. Her vocabulary was not adequate; she used reduced and incomplete sentences, usually three to four words long. She used oppositions of a few concrete adjectives (e.g., small/big, good/bad, black/white). The sentence structure needed more consistency due to the lack of prepositions, pronouns, and adverbs. She used agrammatical forms, and her spontaneous speech was less informational. She was unable to reproduce a short poem with two strophes.

On the Children’s Grammar Test (Table 4), she successfully understood all categories and forms of grammar. She understood and showed adequate answers regarding the plural of nouns, gender, word cases, and the use of pronouns. She was indecisive while answering questions about spatial prepositions (in, on, under, above, in front of, behind, next to, between), adverbs (where, when, how), and terms related to the size of an object (small, smaller, smallest; big, bigger, biggest). The participant answered correctly about plural nouns, irregular forms, all forms of gender, and word cases. She did not use plural forms for pronouns, regardless of whether they were female or male. Her use of spatial prepositions at the expressive level was limited to “in” and “on”, while others she omitted in the sentence or used some phrases to explain the relationship. Similar limitations were present in tasks regarding the size of an object. She could point at the correct answer, but could not answer using the adequate grammatical form. She could use most adverbs during the testing, but a lack of use during her spontaneous speech was evident. Our participant used verbs in the present tense; she did not use past or future tenses. In expressing intention, she used verbs in the present tense with adverbs.

She used short, agrammatical sentences with inadequate word order while forming the narrative based on the Comic Strip Story. She was not able to follow the sequence of events. Her narrative contained informative details but lacked causative relations.

The most noticeable deficit was in her phonetic inventory, which was confirmed on the Global Articulation Test and the Test for Analytical Estimation of Serbian Sounds (Table 5). The analysis showed the quality of speech sounds she could produce and the types of errors made during her production. In the Serbian phonetic inventory, there are 30 phonemes, of which 5 are vowels and 25 are consonants. She correctly produced all five vowels, plosives /p/, /b/, /k/, and /g/, nasals /m/ and /n/, approximant /j/, and fricatives /s/ and /x/. nine consonants were assessed as missing in the form of omission or substitution with another speech sound. She pronounced the other seven speech sounds with some form of distortion. The two plosives, /t/ and /d/, were interdentally pronounced. She omitted plosives in consonant clusters without any pattern, affecting the understanding or semantics of the produced word. She substituted all five sounds in the affricate group (/t͡s/, /t͡ɕ/, /d͡ʑ/, /t͡ʂ/, and /d͡ʐ/) with the sound /k/. The only fricative sound she pronounced adequately in her phonetic inventory was /s/; she used it as a substitute sound for fricatives /z/, /ʂ/, and /ʐ/. She could pronounce the fricative /x/ but did not use it in everyday speech. For the other two fricatives, /v/ and /f/, she used a combination, depending on the sound position in the word, of omission and substitution with the construct /w/, which is not present in the Serbian phonetic inventory. The trill /r/ was omitted, regardless of sound position. The nasal /ɲ/was pronounced distorted, and the lateral /l/ with an interdental position. The lateral /ʎ/ was omitted in her pronunciation. Her pronunciation was characterized by unstable positions that undermined the quality of her speech to varying degrees depending on coarticulation.

The SEPAC scale for sensorimotor development places her achievement at four years and seven months. She was unsuccessful in every activity that included both arms, and she struggled to perform certain actions such as jumping on one leg and standing on one leg with her eyes closed. The Oral Praxis Test showed that out of 31 oral motor models, she successfully imitated 19, partly imitated 4, and could not imitate 8. These oral motor models include movements of face muscles—lips and jaws, and the oral area—tongue and soft palate. She was unsuccessful in almost all models, including those involving tongue movements, and partly successful in models involving lip movements.

## 4. Discussion

The paper presents the findings of the cognitive and speech-language profiles of a 5-year-old girl with APPIS, described as a massive, unilateral, cortical-subcortical stroke localized primarily in the left hemisphere (left MCA territory), after one year of continuous speech-language therapy.

Given the inconsistency of findings concerning the development of a child with perinatal stroke, we aimed to further clarify the findings concerning the cognitive and speech-language development of children with APPIS, with an emphasis on the specificity of each case, where development is influenced not only by the time of the lesion and the degree of damage but also by the child’s neurobiological capacity.

### 4.1. Cognitive Profile

The girl from our study showed uneven cognitive development ranging from below average to average at 5 years old. The participant’s strengths lie within short-term memory, visual perceptual abilities, thinking and planning, good visual short-term memory, as well as the ability to accomplish visual motor tasks within time constraints. Her weaknesses lie within understanding more complex questions and tasks, expressing herself verbally, as well as displaying poor constructive skills.

Although previous studies have demonstrated that a younger age at stroke is a risk factor for less favorable cognitive outcomes [50,51,52,53,54,55], new evidence challenges this assertion by showing that it does not apply linearly across all cognitive outcomes [56,57,58], which was also confirmed in our study. More specifically, many previous studies indicate that children with a history of perinatal unilateral stroke have scores in the normal or near-normal IQ range [28,29,30,69,70,71,72,73]. These results indicate sufficient ongoing plasticity in the developing brain following early focal damage, resulting in the stability and, in some cases, improvement of cognitive functions over time.

Even though findings generally indicate that larger lesion volume [32,57,74,75,76] and a combination of cortical-subcortical strokes are risk factors for negative cognitive outcomes regardless of age at stroke [55,58,74,76], our participant proved them wrong, since her PIQ fell within the normal range.

However, when it comes to verbal and performance IQ scores, the majority of evidence indicates that verbal IQ is most preserved and performance IQ is more affected [77,78,79,80], which is also contradictory to the results obtained in our study.

More specifically, previous studies indicate that perinatal stroke affecting either hemisphere is associated with initial delays in language milestones but an improvement in language ability, with the chance of reaching normal achievement by preschool age [50,81,82,83,84]. Our results showed that the child has persistent language impairment, at preschool age, which is more common in childhood stroke but not in perinatal stroke [52,85].

Furthermore, some researchers [71,86,87,88] have shown that children after perinatal stroke affecting the right hemisphere have difficulties with spatial integration (organizing elements together into a unified whole), whereas left hemisphere lesions result in impairments in processing detail. In our study, it was the opposite: the girl, who had a stroke affecting the left hemisphere, had average results in processing details and trouble with spatial integration.

Our results could be explained by the fact that, in her early development and very young age, the girl had physical therapy, which potentially led to good outcomes and healing [89]. On the other side, she did not start speech therapy until the age of four, which indicates that her development was interrupted, as she missed an opportunity for optimal language learning [90]. This confirms theories that indicate the existence of so-called “sensitive” or “critical” developmental periods.

Also, evidence suggests that lower-order skills, such as simple language, visual, and sensory-motor skills (subsumed by less complex neural networks), often show evidence of good functional recovery. However, more complex skills, which are more likely to be subsumed by complex or diffuse neural networks, have less complete recovery [90]. In the case of our participant, this could refer to higher-order speech and language abilities.

Some authors [34] found that IQ measures did not differ from normative data at preschool age, but emerging deficits were discovered when children were followed longitudinally to school age. Others found decreasing IQ and slowed language development [31,32,33]. However, some studies have demonstrated no change in intelligence measures over time [35,80]. In the case of our participant, follow-up studies need to be done in order to obtain answers to the question of whether her cognitive status will change over time.

As with other investigations of cognition after perinatal stroke, comparisons of these divergent findings are complicated by different diagnostic criteria, mixed populations that include asphyxia and other diffuse injuries [32], as well as the use of different tests (with different normative samples) over different ages. Also, these inconsistent findings may indicate that other moderating variables are most likely at play [34,58,83].

### 4.2. Speech-Language Profile

The child’s evaluated speech-language, sensorimotor, and socio-emotional abilities differed in their stages of development and the child’s chronological age. Speech-language development corresponded to the age of four years and two months, sensory development to four years and seven months, while socio-emotional development was in accordance with the child’s chronological age.

The most notable delay was observed in speech-language development, which lagged ten months behind the chronological age of the child. This finding is consistent with previous research that shows that perinatal stroke, regardless of lateralization, has an impact on language outcomes [50,81,82,83,91]. However, some studies indicate that children with prenatal stroke do not differ in expressive and receptive language scores on neuropsychological testing compared to typically developed children [92].

In general, our case report found early specific differences in language acquisition on comprehension, vocabulary, morphology, use of complex syntax, and narrative structure. The girl was not able to comprehend complex questions. Her vocabulary was insufficiently developed, as she did not use certain types of words such as prepositions, pronouns, adverbs, and auxiliary verbs. She passively adopted many words and grammatical categories, but did not use them actively. During the assessment, she did not answer questions in proper grammatical forms. The sentence she used were shortened and grammatically incorrect. She only used present-tense verbs; she did not use past or future tenses. In terms of narrative abilities, she was able to form narratives based on the Comic Strip Story using short, agrammatical sentences with inadequate word order. She was unable to follow an event’s sequence. Her narrative contained informative details but lacked causal relationships. Furthermore, there was a distinct lack of spontaneous speech.

The present case study’s findings are consistent with the literature pointing to the presence of morphological errors and a reduced mean length of utterance in children with perinatal stroke [51,93,94]. Furthermore, children with left hemisphere perinatal stroke have been reported to have a limited vocabulary [31,81,95]. Moreover, children with prenatal stroke may have difficulties in creating structured narratives, which is reflected in the length of the narrative, the variety of vocabulary used, and the narrative structure [93,96].

In general, speech production abilities have been studied in different categories of children with unilateral perinatal stroke, with growing evidence suggesting that these children have impairments in phonological, syntactical, and semantic processes involved in language production compared to typically developing children [50]. Several studies on language abilities have shown a link between language deficit severity, stroke age, and lesion size and location [27,90,91,97]. Prenatal left-hemispheric lesions are especially analyzed [92,93], considering opportunities for plasticity throughout development. However, despite the facts concerning neuroplasticity, there is evidence that perinatal stroke affects language acquisition and outcomes. Namely, children who sustained left hemispheric brain injury during the prenatal or perinatal period achieve different levels of cognitive and language recovery [90,91]. Additionally, after prenatal or perinatal brain injury, plasticity for some language aspects, especially vocabulary and grammar, may be more substantial than plasticity for other aspects of language, such as narrative skills [51], as observed in the current case study.

When analyzing the present case study, it is essential to consider the girl’s significant progress in speech-language development during continuous one-year speech therapy. Although she still has a speech-language deficit at the age of five, it is important to note her improvement since the beginning of speech-language therapy. This finding indicates that positive stimulation can have a significant impact on developmental functions and associated neural organization during early developmental periods, which are referred to as “critical” or “sensitive” periods of brain development [98]. Furthermore, it suggests that functional plasticity can lead to increased compensation over time, making it crucial to continue therapy and diagnostics with the aim of reaching the child’s full potential.

## 5. Conclusions

The present case study revealed negatively affected cognitive and language outcomes in a five-year-old girl with APPIS localized in the left hemisphere. The girl’s cognitive development has been uneven, ranging from below average to average, and she had specific language acquisition deficits in comprehension, vocabulary, morphology, use of complex syntax, and narrative structure. The obtained results, in terms of cognitive outcomes, present novelty in this area, even when considering the divergent literature data regarding children’s cognitive and speech-language development after perinatal stroke. Furthermore, the obtained results point to the specificity of each child, whose development is influenced not only by the time of the brain lesion and the degree of damage but also by the child’s neurobiological capacity. Therefore, additional studies are required to evaluate the cognitive and speech-language outcomes in children with APPIS to gain a deeper understanding of the individual differences of these children.

Finally, the results highlight the importance of early intervention and systematic monitoring of children with APPIS with the aim of improving their development. It becomes apparent that a multidisciplinary or even interdisciplinary research approach is crucial if we want to better understand the causes and consequences on the child’s overall development. Also, early and comprehensive diagnostics, followed by a targeted individual therapeutic model of work, are prerequisites for achieving optimal results in children with developmental difficulties caused by APPIS.

## Figures and Tables

**Figure 1 children-11-00033-f001:**
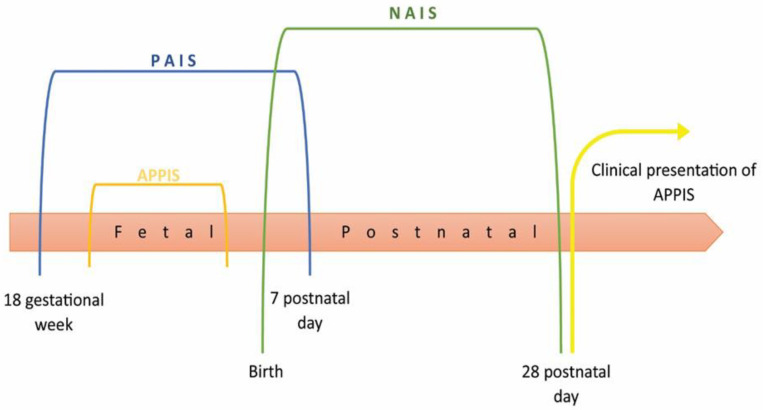
Representation of arterial ischemic stroke terminology based on the time of its occurrence. Legend: PAIS—perinatal arterial ischemic stroke, APPIS—arterial presumed perinatal ischemic stroke, NAIS—neonatal arterial ischemic stroke.

**Figure 2 children-11-00033-f002:**
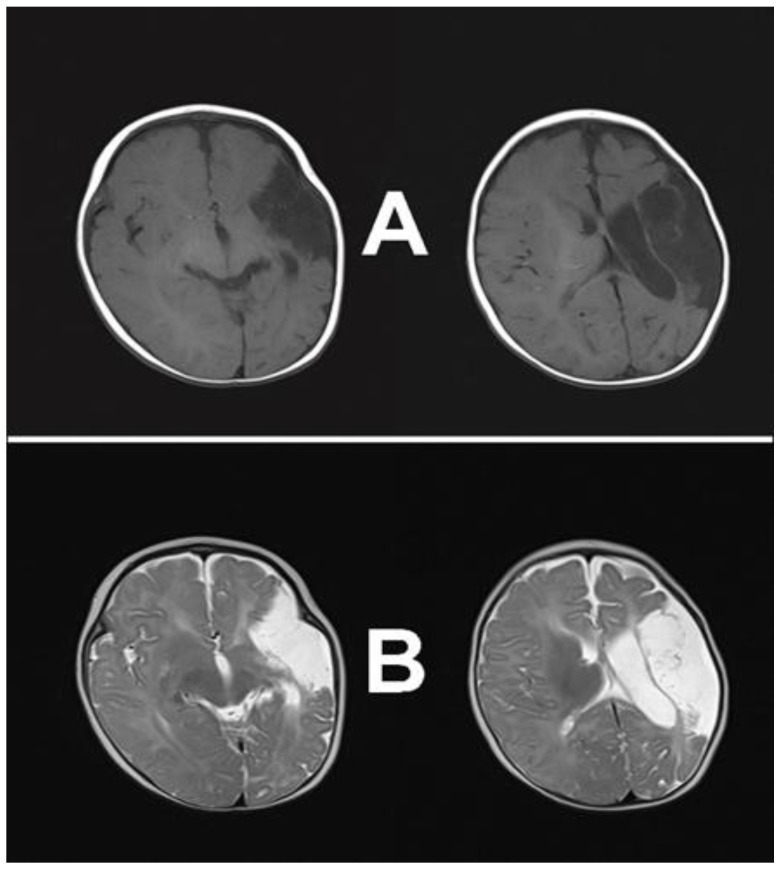
Brain MRI at six months of age. Legend: T_1_ (**A**) and T_2_ (**B**) weighted images.

**Table 1 children-11-00033-t001:** Etiology of perinatal stroke.

Maternal	Antenatal	Intrapartum	Fetal
History of seizures in the family	First pregnancy	Prolonged ruptures of membranes	Male sex
History of neurological diseases	Thrombophilia	Abnormal cardiotocography pattern	Congenital heart disease
	Preeclampsia	Meconium stained amniotic fluid	Hypoglycaemia
	Maternal smoking	Emergency caesarean section	Early onset sepsis/meningitis
		Assisted vacuum delivery	
		Need for resuscitation	
		Apgar scores < 7	
		Birth asphyxia	

**Table 2 children-11-00033-t002:** Results of cognitive assessment.

	Subtests	Score	SD	IQ
Verbal scale	Information	5	0	66
	Comprehension	3	−1	
	Arithmetic	5	0	
	Similarities	4	0	
	Digit Span	6	+1	
Performance scale	Picture Completion	10	0	93
	Picture Arrangement	11	+1	
	Block Design	5	−2	
	Object Assembly	7	−1	
	Coding	12	+2	

**Table 3 children-11-00033-t003:** PPVT and SEPAC results at five years of age.

Instrument	Age Equivalent
PPVT	Six years and one month
SEPAC—speech and language	Four years and two months
SEPAC—sensorimotor	Four years and seven months
SEPAC—socioemotional	Five years

Note. PPVT—Peabody Picture Vocabulary Test; SEPAC—Scale for Evaluation of Psychophysiological Abilities of Children.

**Table 4 children-11-00033-t004:** Results of the Children’s Grammar Test.

	Comprehension of Grammar Forms			Production of Grammar Forms		
	+	+/−	−	+	+/−	−
Plural of nouns	√			√		
Gender	√			√		
Word cases	√			√		
Pronouns	√				√	
Spatial prepositions		√		√ (in, on)		√ (under, above, in front of, behind, next to, between)
Adverbs		√				√
Comparison of adjectives		√				√
Present tense	√			√		
Past tense	√					√
Future tense	√					√

**Table 5 children-11-00033-t005:** Results of Articulation Tests.

			Types of Articulation Errors			
		Target Consonant	Omission	Substitution	Devoiced	Interdental
Plosives	Voiceless	p				
		k				
		t				√
	Voiced	b				
		g			√	
		d			√	√
Affricates	Voiceless	t͡s		k		
		t͡ɕ		k		
		d͡ʑ		k		
	Voiced	t͡ʂ		k		
		d͡ʐ		k		
Fricatives	Voiceless	f	√	w		
		x				
		s				
		z		s		
	Voiced	ʂ		s		
		ʐ		s		
Approximants		ʋ	√	w		
		j				
Nasals		m				
		n				
		ɲ			√	
Laterals		l				√
		ʎ	√			
Trill		r	√			

## Data Availability

Data are contained within the article.

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
