# Peer review of "Arterial Presumed Perinatal Ischemic Stroke: A Mini Review and Case Report of Cognitive and Speech-Language Profiles in a 5-Year-Old Girl"

_children, 2023, doi:10.3390/children11010033_

Round 1

Reviewer 1 Report

Comments and Suggestions for Authors

This case report provides a comprehensive overview of the patient's medical history, diagnosis, and outcomes. The organization is generally clear, with distinct sections covering different aspects such as cognitive profile, speech and language profile, and results.

The detailed medical history and diagnostic process are well-presented. However, it would be helpful to provide a brief summary or conclusion at the end of the medical history section to highlight key points for readers.

The data collection methods and assessment tools are adequately described, providing transparency about the evaluation process.

The cognitive and speech-language profiles are thoroughly presented, supported by specific test scores and observations.

The discussion provides a valuable interpretation of the results in the context of existing literature. It effectively addresses the uniqueness of the case and potential implications for understanding cognitive and speech-language development after perinatal stroke. However, a clearer link between the results and the existing literature would enhance the discussion.

The conclusion succinctly summarizes the key findings and emphasizes the need for additional research. Consider reinforcing the practical implications of the study and potential recommendations for clinicians or researchers working with similar cases.

The language used is generally formal and scientific, appropriate for a case report. However, ensure consistency in terminology and consider defining abbreviations upon first use. Additionally, in some parts, sentences are quite lengthy, and breaking them down for better readability could enhance the overall flow.

While the case report mentions the need for additional studies, providing specific recommendations or hypotheses for future research would strengthen this paper.

The ethical aspects of the study, such as informed consent and adherence to ethical principles, are briefly mentioned. Consider expanding on ethical considerations, particularly concerning the involvement of a pediatric patient, to assure readers of the study's ethical integrity.

The case report provides valuable insights into the cognitive and speech-language outcomes of a child with arterial presumed perinatal ischemic stroke. Addressing the aforementioned points would further enhance the clarity and impact of the report.

Comments on the Quality of English Language

English language is fine.

Author Response

Dear Reviewer,

Please find enclosed the Review Report of our manuscript entitled "Arterial Presumed Perinatal Ischemic Stroke: A Mini Review and Case Report of Cognitive and Speech-Language Profile in a 5-Year-Old Girl".  

Thank you for recognizing the significance of research and practical issues concerning the cognitive, speech, and language profiles of children who have experienced a perinatal stroke. This knowledge might be particularly useful in the field of speech-language pathology and the overall rehabilitation of affected children.

We are very grateful for your extremely valuable comments: we have modified the paper according to your suggestions.

We sincerely hope that the revised version of the manuscript will receive a positive further review and that it will be accepted for publication in the Children.

Reviewer 2 Report

Comments and Suggestions for Authors

Dear Authors,

The aim of the study was to give a thorough account of the early perinatal ischemic local brain stroke case. The issue of the early perinatal stroke cognitive outcome is very pertinent and has not yet received much attention. The 5-year-old girl's cognitive and speech-language impairments were the main focus of the study. However, it should be mentioned that the authors failed to explain the purpose and relevance of the study in the manuscript.

The title of the manuscript states that the primary focus of the assessment was on language and cognitive development, but the introduction provides very little information from the scientific literature regarding language and cognitive development following perinatal stroke.

From a methodological perspective, it appears improper to use REVISK for an intelligence assessment of a child with severe language disorder because the child is required to understand verbal instructions and provide verbal arguments in the five subtests of the verbal scale. The verbal intelligence index will not be valid in this context. In this instance, the Raven’s color matrices appear more pertinent and reliable.

The tests used for language assessment should be described more extensively with comments about Serbian. Without it, readers will have no possibility to properly understand the results of the language assessment.

There should be a more thorough description of the language assessment tests, including comments regarding Serbian language. Otherwise, readers will not be able to comprehend the results of language assessment.

The result section lacks organization. The quantitative results of the language assessment should be presented in a table format. It is not clear why the SD is presented in the table with REVISK results. What does SD mean in this particular assessment?

In the passages devoted to speech and language, no quantitative data is presented. In this regard, there are no reasons to discuss it as the ‘Speech and language profile’. These data are primarily descriptive and do not provide much new scientific information.

Overall conclusions: the manuscript does not fit in its current state. I would recommend major revision and a second round of review.

Author Response

Dear Reviewer,

Please find enclosed the Review Report of our manuscript entitled "Arterial Presumed Perinatal Ischemic Stroke: A Mini Review and Case Report of Cognitive and Speech-Language Profile in a 5-Year-Old Girl"

 Thank you for acknowledging the study and its very important topic. We are very grateful for your and other reviewer´s valuable comments and suggestions. We have modified the manuscript according to the suggestions.

We sincerely hope that the revised version of the manuscript will receive a positive further review and that it will be accepted for publication in the Children.

Sincerely,

Ljiljana Jeličić

Reviewer 3 Report

Comments and Suggestions for Authors

The abstract of the article is well presented, accurately reflecting the content of the article.

The introduction is well structured covering the topic of the article very informatively.

In the first part of the presented material, a relatively short but meaningful overview of the topic is made.The overview of the topic is not sufficiently controversial to bring out the purpose of analysis of the presented case.

The presented case is analyzed in detail. Neuroimaging methods, combined neurophysiological and a battery of psychological tests were used. All of them are described in detail and clearly.

The case is presented comprehensibly and thoroughly.

The discussion is detailed and informative.

The conclusion is tight and clear

The Reviewer

Author Response

Dear Reviewer,

Please find enclosed the Review Report of our manuscript entitled "Arterial Presumed Perinatal Ischemic Stroke: A Mini Review and Case Report of Cognitive and Speech-Language Profile in a 5-Year-Old Girl"

Thank you for recognizing the significance of research and practical issues related to the cognitive and speech-language profiles of children who have experienced perinatal insult. This understanding could be particularly beneficial for the field of speech-language pathology in addition to the general rehabilitation of these children.

We are very grateful for your and other reviewer´s valuable comments and suggestions. We have modified the manuscript according to the suggestions.

We sincerely hope that the revised version of the manuscript will receive a positive further review and that it will be accepted for publication in the Children.  

Sincerely,

Ljiljana Jeličić

Round 2

Reviewer 2 Report

Comments and Suggestions for Authors

Dear authors,

Upon reviewing your modified content, I have determined that very few alterations were made. You failed to take into account all seven aspects of my comments. The manuscript is currently unsuitable in this regard. I suggest conducting a significant revision and conducting a second round of evaluation.

My comments from the 1st review:

  1. ‘’The aim of the study was is to give a thorough account of the early perinatal ischemic local brain stroke case. The issue of the early perinatal stroke cognitive outcome is very pertinent and has not yet received much attention. The 5-year-old girl's cognitive and speech-language impairments were the main focus of the study. However, it should be mentioned that the authors failed to explain the purpose and relevance of the study in the manuscript.’’

It was suggested that you would provide additional text explaining the goal and significance of the study. No alterations were made to the manuscript.

2.’’The title of the manuscript states that the primary focus of the assessment was on language and cognitive development, but the introduction provides very little information from the scientific literature regarding language and cognitive development following perinatal stroke.’’

The manuscript should include pertinent material, as was suggested. No alterations were made to the manuscript.

3.’’From a methodological perspective, it appears improper to use REVISK for an intelligence assessment of a child with severe language disorder because the child is required to understand verbal instructions and provide verbal arguments in the five subtests of the verbal scale. The verbal intelligence index will not be valid in this context. In this instance, the Raven’s color matrices appear more pertinent and reliable.’’

A significant limitation of the study is that the REVISK was utilized to test intelligence in individuals with restricted speech and language capabilities.

4.’’The tests used for language assessment should be described more extensively with comments about Serbian. Without it, readers will have no possibility to properly understand the results of the language assessment.’’

The primary focus of the case study is on the speech and language problem. However, most readers from countries with different languages have little knowledge about the specific characteristics of the Serbian language and are unable to connect your data with similar occurrences in other languages.

5.’’There should be a more thorough description of the language assessment tests, including comments regarding Serbian language. Otherwise, readers will not be able to comprehend the results of language assessment.’’

See comment to previous point. Nothing was changed in the manuscript.

7.’’In the passages devoted to speech and language, no quantitative data is presented. In this regard, there are no reasons to discuss it as the ‘Speech and language profile’. These data are primarily descriptive and do not provide much new scientific information.’’

I suggest incorporating qualitative data into the sections dedicated to speech and language. However, you overlooked this comment.

Author Response

Dear Reviewer,

Please find enclosed the review report (round II).

Kind regards,

Ljiljana Jeličić

Round 3

Reviewer 2 Report

Comments and Suggestions for Authors

No comments